# Approximate Solutions of the Damped Wave Equation and Dissipative Wave Equation in Fractal Strings

**Dumitru Baleanu [1,2,*]** and **Hassan Kamil Jassim [3]**

1   Department of Mathematics, Faculty of Arts and Sciences, Cankaya University, 06530 Ankara, Turkey
2   Institute of Space Sciences, Magurele, P.O.Box MG-23, Ro 077125 Bucharest, Romania
3   Department of Mathematicas, Faculty of Education for Pure Sciences, University of Thi Qar, Nasiriyah 64001, Iraq; hassan.kamil@yahoo.com
*   Correspondence: dumitru@cankaya.edu.tr

**Abstract:** In this paper, we apply the local fractional Laplace variational iteration method (LFLVIM) and the local fractional Laplace decomposition method (LFLDM) to obtain approximate solutions for solving the damped wave equation and dissipative wave equation within local fractional derivative operators (LFDOs). The efficiency of the considered methods are illustrated by some examples. The results obtained by LFLVIM and LFLDM are compared with the results obtained by LFVIM. The results reveal that the suggested algorithms are very effective and simple, and can be applied for linear and nonlinear problems in sciences and engineering.

**Keywords:** LFDOs; LFLVIM; LFLDM; damped wave equation; dissipative wave equation

## 1. Introduction

The local fractional calculus was successfully utilized to describe the non-differentiable problems arising in mathematical physics, such as the diffusion equations [1–4], the gas dynamic equation [5], the telegraph equation [6], wave equation [7], Fokker Planck equation [8,9], Laplace equation [10], Klein–Gordon equations [11,12], Helmholtz equation [13,14], Goursat Problem [15] and other differential equations [16–18] on Cantor sets. The existence and uniqueness of solutions for local fractional differential equations [19,20]. Recently, the dissipative wave equation with LFDOs was given by [21]:

$$\frac{\partial^{2\vartheta}\varphi(\eta,\kappa)}{\partial\kappa^{2\vartheta}} - \frac{\partial^{\vartheta}\varphi(\eta,\kappa)}{\partial\kappa^{\vartheta}} - \frac{\partial^{2\vartheta}\varphi(\eta,\kappa)}{\partial\eta^{2\vartheta}} - \frac{\partial^{\vartheta}\varphi(\eta,\kappa)}{\partial\eta^{\vartheta}} - f_1(\eta,\kappa) = 0, \quad 0 < \vartheta \leq 1, \tag{1}$$

$$\varphi(\eta,0) = \psi_1(\eta), \quad \frac{\partial^{\vartheta}\varphi(\eta,0)}{\partial\kappa^{\vartheta}} = \psi_2(\eta),$$

as well as the damped wave equation with LFDOs was given by [21]:

$$\frac{\partial^{2\vartheta}\varphi(\eta,\kappa)}{\partial\kappa^{2\vartheta}} - \frac{\partial^{\vartheta}\varphi(\eta,\kappa)}{\partial\kappa^{\vartheta}} - \frac{\partial^{2\vartheta}\varphi(\eta,\kappa)}{\partial\eta^{2\vartheta}} - f_2(\eta,\kappa) = 0, \quad 0 < \vartheta \leq 1, \tag{2}$$

$$\varphi(\eta,0) = \phi_1(\eta), \quad \frac{\partial^{\vartheta}\varphi(\eta,0)}{\partial\kappa^{\vartheta}} = \phi_2(\eta).$$

The authors in [21] proposed the LFVIM to consider the dissipative wave equation and the damped wave equation with LFDOs. The main target of this paper is to use LFLVIM and LFLDM to implement the dissipative wave equation and the damped wave equation in fractal strings.

The theory of fractal strings has been developed over the past years by Lapidus and co-workers in a series of papers [22,23]. A standard fractal string is a bounded open subset of the real number line. Such a set is a disjoint union of open intervals, the lengths of which form a sequence $J = \{\ell_i\}_{i=1}^{\infty}$ which we typically assume to be infinite [23].

In recent years, a variety of numerical and analytical methods have been applied to solve the PDEs with LFDOs, such as local fractional differential transform method [24], local fractional series expansion method [25], local fractional reduce differential transform method [26] and other methods [27,28].The paper has been organized as follows. In Section 2, the basic mathematical tools are reviewed. In Section 3, we give analysis of the methods used. In Section 4, we consider two illustrative examples. Finally, in Section 5, we present our conclusions.

## 2. Analysis of LFLVIM

Let us consider the following local fractional partial differential equation with LFDOs:

$$L_\vartheta \varphi(\eta, \kappa) + R_\vartheta \varphi(\eta, \kappa) + N_\vartheta \varphi(\eta, \kappa) = \omega(\eta, \kappa), \tag{3}$$

where $L_\vartheta = \dfrac{\partial^{m\vartheta}}{\partial \kappa^{m\vartheta}}$ denotes the linear LFDO, $R_\vartheta$ is the remaining linear operator, $N_\vartheta$ represents the general nonlinear differential operator, and $\omega$ is the source term.

According tothe rule of LFVIM [27–29]:

$$\varphi_{n+1}(\kappa) = \varphi_n(\kappa) + {}_0 I_\kappa^{(\vartheta)} \left[ \sigma(\kappa - \xi)^\vartheta \left( L_\vartheta \left[ \varphi_n(\xi) \right] + R_\vartheta \left[ \widetilde{\varphi}_n(\xi) \right] + N_\vartheta \left[ \widetilde{\varphi}_n(\xi) \right] - \omega(\xi) \right) \right], \tag{4}$$

where $\sigma(\kappa - \xi)^\vartheta$ is a fractal Lagrange multiplier.

For the initial value problems of (3), we can start with:

$$u_0(\eta, \kappa) = u(\eta, 0) + \frac{\kappa^\vartheta}{\Gamma(1+\vartheta)} u^{(\vartheta)}(\eta, 0) + \cdots + \frac{\kappa^{(m-1)\vartheta}}{\Gamma(1+(m-1)\vartheta)} u^{((m-1)\vartheta)}(\eta, 0). \tag{5}$$

We now take local fractional Laplace transform for (4), we get:

$$\begin{aligned} \widetilde{L}_\vartheta \left\{ \varphi_{n+1}(\kappa) \right\} &= \widetilde{L}_\vartheta \left\{ \varphi_n(\kappa) \right\} + \widetilde{L}_\vartheta \left\{ \sigma(\kappa)^\vartheta \right\} \times \\ &\quad \widetilde{L}_\vartheta \left\{ L_\vartheta \left[ \varphi_n(\xi) \right] + R_\vartheta \left[ \widetilde{\varphi}_n(\xi) \right] + N_\vartheta \left[ \widetilde{\varphi}_n(\xi) \right] - \omega(\xi) \right\}. \end{aligned} \tag{6}$$

Taking the LF variation of (6):

$$\begin{aligned} \delta^\vartheta \left( \widetilde{L}_\vartheta \left\{ \varphi_{n+1}(\kappa) \right\} \right) &= \delta^\vartheta \left( \widetilde{L}_\vartheta \left\{ \varphi_n(\kappa) \right\} \right) + \\ \delta^\vartheta \left( \widetilde{L}_\vartheta \left\{ \sigma(\kappa)^\vartheta \right\} \widetilde{L}_\vartheta \left\{ (L_\vartheta \left[ \varphi_n(\kappa) \right] + R_\vartheta \left[ \widetilde{\varphi}_n(\kappa) \right] + N_\vartheta \left[ \widetilde{\varphi}_n(\kappa) \right] - \omega(\kappa)) \right\} \right). \end{aligned} \tag{7}$$

By using computation of (7), we get:

$$\begin{aligned} \delta^\vartheta \left( \widetilde{L}_\vartheta \left\{ \varphi_{n+1}(\kappa) \right\} \right) &= \delta^\vartheta \left( \widetilde{L}_\vartheta \left\{ \varphi_n(\kappa) \right\} \right) + \widetilde{L}_\alpha \left\{ \sigma(\kappa)^\vartheta \right\} \delta^\vartheta \left( \widetilde{L}_\vartheta \left\{ L_\vartheta \left[ \varphi_n(\kappa) \right] \right\} \right) \\ &= 0. \end{aligned} \tag{8}$$

Hence, from (8) we get:

$$1 + \widetilde{L}_\vartheta \left\{ \sigma(\kappa)^\vartheta \right\} s^{m\vartheta} = 0, \tag{9}$$

where

$$
\begin{aligned}
\delta^\vartheta \left( \widetilde{L}_\vartheta \left\{ L_\vartheta \left[ \varphi_n(\kappa) \right] \right\} \right) &= \delta^\vartheta \left( s^{m\vartheta} \widetilde{L}_\vartheta \left\{ \varphi_n(\kappa) \right\} - s^{(m-1)\vartheta} \varphi_n(0) - \cdots - \varphi_n^{((m-1)\vartheta)}(0) \right) \\
&= s^{m\vartheta} \delta^\vartheta \left( \widetilde{L}_\vartheta \left\{ \varphi_n(\kappa) \right\} \right).
\end{aligned}
\tag{10}
$$

Therefore, we have

$$
\widetilde{L}_\vartheta \left\{ \sigma(\kappa)^\vartheta \right\} = -\frac{1}{s^{m\vartheta}}.
\tag{11}
$$

Taking the inverse of into (11), we obtain:

$$
\sigma(\kappa)^\vartheta = \widetilde{L}_\vartheta^{-1} \left( -\frac{1}{s^{m\vartheta}} \right) = -\frac{\kappa^{(m-1)\vartheta}}{\Gamma(1 + (m-1)\vartheta)}.
\tag{12}
$$

Hence, we have the following iteration algorithm:

$$
\begin{aligned}
\widetilde{L}_\vartheta \left\{ \varphi_{n+1}(\kappa) \right\} &= \widetilde{L}_\vartheta \left\{ \varphi_n(\kappa) \right\} - \frac{1}{s^{m\vartheta}} \widetilde{L}_\vartheta \left\{ L_\vartheta \left[ \varphi_n(\kappa) \right] + R_\vartheta \left[ \varphi_n(\kappa) \right] + N_\vartheta \left[ \varphi_n(\kappa) \right] - \omega(\kappa) \right\} \\
&= \widetilde{L}_\vartheta \left\{ \varphi_n(\kappa) \right\} - \frac{1}{s^{m\vartheta}} \widetilde{L}_\vartheta \left\{ s^{m\vartheta} \varphi_n(\kappa) - \cdots - \varphi_n^{((m-1)\vartheta)}(0) \right\} \\
&\quad - \frac{1}{s^{m\vartheta}} \widetilde{L}_\vartheta \left\{ R_\vartheta \left[ \varphi_n(\kappa) \right] + N_\vartheta \left[ \varphi_n(\kappa) \right] - \omega(\kappa) \right\} \\
&= \frac{1}{s^\vartheta} \varphi_n(0) - \frac{1}{s^{2\vartheta}} \varphi_n^{(\vartheta)}(0) - \cdots - \frac{1}{s^{m\vartheta}} \varphi_n^{((m-1)\vartheta)}(0) \\
&\quad - \frac{1}{s^{m\vartheta}} \widetilde{L}_\vartheta \left\{ R_\vartheta \left[ \varphi_n(\kappa) \right] + N_\vartheta \left[ \varphi_n(\kappa) \right] - \omega(\kappa) \right\}.
\end{aligned}
\tag{13}
$$

Therefore, the solution of (3) is

$$
\varphi(\eta, \kappa) = \lim_{n \to \infty} \widetilde{L}_\vartheta^{-1} \left( \widetilde{L}_\vartheta \left\{ \varphi_n(\eta, \kappa) \right\} \right).
\tag{14}
$$

## 3. Analysis of the Local Fractional Laplace Decomposition Method

We now consider the local fractionaloperator equation in the following form:

$$
L_\vartheta \varphi(\eta, \kappa) + R_\vartheta \varphi(\eta, \kappa) = h(\eta, \kappa).
\tag{15}
$$

Taking LFLT on (15), we obtain

$$
\widetilde{L}_\vartheta \left\{ L_\vartheta \varphi(\eta, \kappa) \right\} + \widetilde{L}_\vartheta \left\{ R_\vartheta \varphi(\eta, \kappa) \right\} = \widetilde{L}_\vartheta \left\{ h(\eta, \kappa) \right\}.
\tag{16}
$$

By applying the LFLT differentiation property, we have

$$
\begin{aligned}
s^{m\vartheta} \widetilde{L}_\vartheta \left\{ \varphi(\eta, \kappa) \right\} &- s^{(m-1)\vartheta} \varphi(\eta, 0) - s^{(m-2)\vartheta} \varphi(\eta, 0) - \cdots - u^{((m-1)\vartheta)}(\eta, 0) \\
&+ \widetilde{L}_\vartheta \{ R_\alpha \varphi(\eta, \kappa) \} = \widetilde{L}_\vartheta \{ h(\eta, \kappa) \},
\end{aligned}
\tag{17}
$$

or equivalently

$$
\begin{aligned}
\widetilde{L}_\vartheta \left\{ \varphi(\kappa) \right\} &= \frac{1}{s^\vartheta} \varphi(0) + \frac{1}{s^{2\vartheta}} \varphi^{(\vartheta)}(0) + \cdots + \frac{1}{s^{m\vartheta}} \varphi^{((m-1)\vartheta)}(0) \\
&\quad + \frac{1}{s^{m\vartheta}} \widetilde{L}_\vartheta \left\{ h(\kappa) \right\} - \frac{1}{s^{m\vartheta}} \widetilde{L}_\vartheta \left\{ R_\vartheta \varphi(\kappa) \right\}.
\end{aligned}
\tag{18}
$$

Taking the inverse of LFLT on (18), we have

$$
\begin{aligned}
\varphi(\kappa) \;=\;& \varphi(0) + \cdots + \frac{\kappa^{(m-1)\vartheta}}{\Gamma(1+(m-1)\vartheta)}\varphi^{(\vartheta)}(0) \\
&+ \widetilde{L}_\vartheta^{-1}\left(\frac{1}{s^{m\vartheta}}\widetilde{L}_\vartheta\{h(\eta,\kappa)\}\right) - \widetilde{L}_\vartheta^{-1}\left(\frac{1}{s^{m\vartheta}}\widetilde{L}_\vartheta\{R_\vartheta\varphi(\eta,\kappa)\}\right).
\end{aligned}
\tag{19}
$$

We are going to represent the solution in an infinite series given below:

$$
\varphi(\eta,\kappa) = \sum_{n=0}^{\infty} \varphi_n(\eta,\kappa).
\tag{20}
$$

Substituting (20) into (19), which gives us this result

$$
\begin{aligned}
&\sum_{n=0}^{\infty} \varphi_n(\eta,\kappa) \\
&= \varphi(\eta,0) + \cdots + \widetilde{L}^{-1}\left(\frac{1}{s^{m\vartheta}}L_\vartheta\{h(\eta,\kappa)\}\right) - \widetilde{L}_\vartheta^{-1}\left(\frac{1}{s^{m\vartheta}}\widetilde{L}\left\{R_\vartheta\sum_{n=0}^{\infty}\varphi_n(\eta,\kappa)\right\}\right).
\end{aligned}
\tag{21}
$$

When we compare the left and right hand sides of (21) we obtain

$$
\varphi_0(\eta,\kappa) \;=\; \varphi(\eta,0) + \frac{\kappa^\vartheta}{\Gamma(1+\vartheta)}\varphi^{(\vartheta)}(\eta,0) + \cdots + \widetilde{L}^{-1}\left(\frac{1}{s^{m\vartheta}}\widetilde{L}_\vartheta\{h(\eta,\kappa)\}\right),
\tag{22}
$$

$$
\varphi_{n+1}(\eta,\kappa) \;=\; -\widetilde{L}_\vartheta^{-1}\left(\frac{1}{s^{m\vartheta}}\widetilde{L}_\vartheta\{R_\vartheta\varphi_n(\eta,\kappa)\}\right).
\tag{23}
$$

## 4. Illustrative Examples

**Example 1.** *Consider dissipative wave equation with local fractional derivative operators:*

$$
\frac{\partial^{2\vartheta}\varphi}{\partial\kappa^{2\vartheta}} - \frac{\partial^\vartheta\varphi}{\partial\kappa^\vartheta} - \frac{\partial^{2\vartheta}\varphi}{\partial\eta^{2\vartheta}} - \frac{\partial^\vartheta\varphi}{\partial\eta^\vartheta} - \frac{\kappa^\vartheta}{\Gamma(1+\vartheta)} = 0, \quad 0 < \vartheta \le 1,
\tag{24}
$$

*with the initial conditions:*

$$
\varphi(\eta,0) = \frac{\eta^\vartheta}{\Gamma(1+\vartheta)}, \quad \frac{\partial^\vartheta\varphi(\eta,0)}{\partial\kappa^\vartheta} = 0.
\tag{25}
$$

*Now,*

$$
\begin{aligned}
\widetilde{L}_\vartheta\{\varphi_{n+1}(\eta,\kappa)\} \;=\;& \widetilde{L}_\vartheta\{\varphi_n(\eta,\kappa)\} - \\
& \frac{1}{s^{2\vartheta}}\widetilde{L}_\vartheta\left\{\frac{\partial^{2\vartheta}\varphi_n}{\partial\kappa^{2\vartheta}} - \frac{\partial^\vartheta\varphi_n}{\partial\kappa^\vartheta} - \frac{\partial^{2\vartheta}\varphi_n}{\partial\eta^{2\vartheta}} - \frac{\partial^\vartheta\varphi_n}{\partial\eta^\vartheta} - \frac{\kappa^\vartheta}{\Gamma(1+\vartheta)}\right\} \\
\;=\;& \widetilde{L}_\vartheta\{\varphi_n(\eta,\kappa)\} - \frac{1}{s^{2\vartheta}}\left(s^{2\vartheta}\widetilde{L}_\vartheta\{\varphi_n(\eta,\kappa)\} - s^\vartheta\varphi_n(\eta,0) - \varphi_n^{(\vartheta)}(\eta,0)\right) + \\
& \frac{1}{s^{2\vartheta}}\widetilde{L}_\vartheta\left\{\frac{\partial^{2\vartheta}\varphi_n(\eta,\kappa)}{\partial\eta^{2\vartheta}} + \frac{\partial^\vartheta\varphi_n(\eta,\kappa)}{\partial\eta^\vartheta} + \frac{\partial^\vartheta\varphi_n(\eta,\kappa)}{\partial\kappa^\vartheta} + \frac{\kappa^\vartheta}{\Gamma(1+\vartheta)}\right\} \\
\;=\;& \frac{1}{s^\vartheta}\varphi_n(\eta,0) + \frac{1}{s^{2\vartheta}}\varphi_n^{(\vartheta)}(\eta,0) + \frac{1}{s^{4\vartheta}} + \\
& \frac{1}{s^{2\vartheta}}\widetilde{L}_\vartheta\left\{\frac{\partial^{2\vartheta}\varphi_n(\eta,\kappa)}{\partial\eta^{2\vartheta}} + \frac{\partial^\vartheta\varphi_n(\eta,\kappa)}{\partial\eta^\vartheta} + \frac{\partial^\vartheta\varphi_n(\eta,\kappa)}{\partial\kappa^\vartheta}\right\}.
\end{aligned}
\tag{26}
$$

*The initial value reads:*

$$
\varphi_0(\eta,\kappa) = \frac{\eta^\vartheta}{\Gamma(1+\vartheta)}.
\tag{27}
$$

*Hence, we get the first approximation, namely:*

$$
\begin{aligned}
\widetilde{L}_\vartheta\left\{\varphi_1(\eta,\kappa)\right\} &= \frac{1}{s^\vartheta}\varphi_0(\eta,0) + \frac{1}{s^{2\vartheta}}\varphi_0^{(\vartheta)}(\eta,0) + \frac{1}{s^{4\vartheta}} + \\
&\quad \frac{1}{s^{2\vartheta}}\widetilde{L}_\vartheta\left\{\frac{\partial^{2\vartheta}\varphi_0(\eta,\kappa)}{\partial\eta^{2\vartheta}} + \frac{\partial^\vartheta\varphi_0(\eta,\kappa)}{\partial\eta^\vartheta} + \frac{\partial^\vartheta\varphi_0(\eta,\kappa)}{\partial\kappa^\vartheta}\right\} \\
&= \frac{1}{s^\vartheta}\frac{\eta^\vartheta}{\Gamma(1+\vartheta)} + \frac{1}{s^{3\vartheta}} + \frac{1}{s^{4\vartheta}}.
\end{aligned}
$$

*Thus, we have*

$$
\varphi_1(\eta,\kappa) = \widetilde{L}_\vartheta^{-1}\left(\frac{1}{s^\vartheta}\frac{\eta^\vartheta}{\Gamma(1+\vartheta)} + \frac{1}{s^{3\vartheta}} + \frac{1}{s^{4\vartheta}}\right).
$$

*The second approximations reads:*

$$
\begin{aligned}
\widetilde{L}_\vartheta\left\{\varphi_2(\eta,\kappa)\right\} &= \frac{1}{s^\vartheta}\varphi_1(\eta,0) + \frac{1}{s^{2\vartheta}}\varphi_1^{(\vartheta)}(\eta,0) + \frac{1}{s^{4\vartheta}} + \\
&\quad \frac{1}{s^{2\vartheta}}\widetilde{L}_\vartheta\left\{\frac{\partial^{2\vartheta}\varphi_1(\eta,\kappa)}{\partial\eta^{2\vartheta}} + \frac{\partial^\vartheta\varphi_1(\eta,\kappa)}{\partial\eta^\vartheta} + \frac{\partial^\vartheta\varphi_1(\eta,\kappa)}{\partial\kappa^\vartheta}\right\} \\
&= \frac{1}{s^\vartheta}\frac{\eta^\vartheta}{\Gamma(1+\vartheta)} + \frac{1}{s^{3\vartheta}} + \frac{1}{s^{4\vartheta}} + \frac{1}{s^{4\vartheta}} + \frac{1}{s^{5\vartheta}}.
\end{aligned}
$$

*Therefore, we get*

$$
\varphi_2(\eta,\kappa) = \widetilde{L}_\vartheta^{-1}\left(\frac{1}{s^\vartheta}\frac{\eta^\vartheta}{\Gamma(1+\vartheta)} + \frac{1}{s^{3\vartheta}} + \frac{1}{s^{4\vartheta}} + \frac{1}{s^{4\vartheta}} + \frac{1}{s^{4\vartheta}} + \frac{1}{s^{5\vartheta}}\right).
$$

*The other approximations are written as:*

$$
\begin{aligned}
\widetilde{L}_\vartheta\left\{\varphi_3(\eta,\kappa)\right\} &= \frac{1}{s^\vartheta}\varphi_2(\eta,0) + \frac{1}{s^{2\vartheta}}\varphi_2^{(\vartheta)}(\eta,0) + \frac{1}{s^{4\vartheta}} + \\
&\quad \frac{1}{s^{2\vartheta}}\widetilde{L}_\vartheta\left\{\frac{\partial^{2\vartheta}\varphi_2(\eta,\kappa)}{\partial\eta^{2\vartheta}} + \frac{\partial^\vartheta\varphi_2(\eta,\kappa)}{\partial\eta^\vartheta} + \frac{\partial^\vartheta\varphi_2(\eta,\kappa)}{\partial\kappa^\vartheta}\right\} \\
&= \frac{1}{s^\vartheta}\frac{\eta^\vartheta}{\Gamma(1+\vartheta)} + \frac{1}{s^{3\vartheta}} + \frac{1}{s^{4\vartheta}} + \frac{1}{s^{4\vartheta}} + \frac{1}{s^{5\vartheta}} + \frac{1}{s^{5\vartheta}} + \frac{1}{s^{6\vartheta}}.
\end{aligned}
$$

*Therefore, we have*

$$
\varphi_3(\eta,\kappa) = \widetilde{L}_\vartheta^{-1}\left(\frac{1}{s^\vartheta}\frac{\eta^\vartheta}{\Gamma(1+\vartheta)} + \frac{1}{s^{3\vartheta}} + \frac{1}{s^{4\vartheta}} + \frac{1}{s^{4\vartheta}} + \frac{1}{s^{5\vartheta}} + \frac{1}{s^{5\vartheta}} + \frac{1}{s^{6\vartheta}}\right).
$$

*The same manner, we get*

$$
\begin{aligned}
\widetilde{L}_\vartheta\left\{\varphi_n(\eta,\kappa)\right\} &= \frac{1}{s^{2\vartheta}}\widetilde{L}_\vartheta\left\{\frac{\partial^{2\vartheta}\varphi_{n-1}(\eta,\kappa)}{\partial\eta^{2\vartheta}} + \frac{\partial^\vartheta\varphi_{n-1}(\eta,\kappa)}{\partial\eta^\vartheta} + \frac{\partial^\vartheta\varphi_{n-1}(\eta,\kappa)}{\partial\kappa^\vartheta}\right\} \\
&= \frac{1}{s^\vartheta}\frac{\eta^\vartheta}{\Gamma(1+\vartheta)} + \left[\frac{1}{s^{3\vartheta}} + \frac{1}{s^{4\vartheta}} + \frac{1}{s^{5\vartheta}} + \cdots + \frac{1}{s^{(n+2)\vartheta}}\right] + \\
&\quad \left[\frac{1}{s^{4\vartheta}} + \frac{1}{s^{5\vartheta}} + \cdots + \frac{1}{s^{(n+3)\vartheta}}\right].
\end{aligned}
$$

*Consequently, the local fractional series solution is:*

$$
\begin{aligned}
\varphi(\eta,\kappa) &= \lim_{n\to\infty} \widetilde{L}_\vartheta^{-1}\left(\widetilde{L}_\vartheta\{\varphi_n(\eta,\kappa)\}\right)\\
&= \frac{\eta^\vartheta}{\Gamma(1+\vartheta)} + 2E_\vartheta(\kappa^\vartheta) - \sum_{r=0}^{1}\frac{\kappa^{r\vartheta}}{\Gamma(1+r\vartheta)} - \sum_{r=0}^{2}\frac{\kappa^{r\vartheta}}{\Gamma(1+r\vartheta)},
\end{aligned}
\tag{28}
$$

*which is exactly the same as that obtained by LFVIM [16].*

*Now, we solve problem* (24) *by using the LFLDM. From* (22)–(25), *the iteration algorithm can be written as follows:*

$$
\varphi_0 = \frac{\eta^\alpha}{\Gamma(1+\vartheta)} + \frac{\kappa^{3\vartheta}}{\Gamma(1+3\vartheta)},
\tag{29}
$$

$$
\varphi_{n+1} = \widetilde{L}_\vartheta^{-1}\left[\frac{1}{s^{2\vartheta}}\widetilde{L}_\vartheta\left\{\frac{\partial^{2\vartheta}\varphi_n(\eta,\kappa)}{\partial\eta^{2\vartheta}} + \frac{\partial^\vartheta\varphi_n(\eta,\kappa)}{\partial\eta^\vartheta} + \frac{\partial^\vartheta\varphi_n(\eta,\kappa)}{\partial\kappa^\vartheta}\right\}\right].
\tag{30}
$$

*Therefore, from* (29) *and* (30) *we give the components as follows:*

$$
\begin{aligned}
\varphi_1 &= \widetilde{L}_\vartheta^{-1}\left[\frac{1}{s^{2\vartheta}}\widetilde{L}_\vartheta\left\{\frac{\partial^{2\vartheta}\varphi_0}{\partial\eta^{2\vartheta}} + \frac{\partial^\vartheta\varphi_0}{\partial\eta^\vartheta} + \frac{\partial^\vartheta\varphi_0}{\partial\kappa^\vartheta}\right\}\right]\\
&= \widetilde{L}_\vartheta^{-1}\left(\frac{1}{s^{3\vartheta}} + \frac{1}{s^{5\vartheta}}\right),
\end{aligned}
$$

$$
\begin{aligned}
\varphi_2 &= \widetilde{L}_\vartheta^{-1}\left[\frac{1}{s^{2\vartheta}}\widetilde{L}_\vartheta\left\{\frac{\partial^{2\vartheta}\varphi_1}{\partial\eta^{2\vartheta}} + \frac{\partial^\vartheta\varphi_1}{\partial\eta^\vartheta} + \frac{\partial^\vartheta\varphi_1}{\partial\kappa^\vartheta}\right\}\right]\\
&= \widetilde{L}_\vartheta^{-1}\left(\frac{1}{s^{4\vartheta}} + \frac{1}{s^{6\vartheta}}\right).
\end{aligned}
$$

$$\vdots$$

*Consequently, we obtain*

$$
\varphi(\eta,\kappa) = \frac{\eta^\vartheta}{\Gamma(1+\vartheta)} + 2E_\vartheta(\kappa^\vartheta) - \sum_{r=0}^{1}\frac{\kappa^{r\vartheta}}{\Gamma(1+r\vartheta)} - \sum_{r=0}^{2}\frac{\kappa^{r\vartheta}}{\Gamma(1+r\vartheta)},
\tag{31}
$$

*which is exactly the same as that obtained by LFLVIM and LFVIM [21].*

In Figures 1 and 2, the 3-dimensional plots of the approximate solutions of (24) with initialcondition (25) are shown for different values of $\vartheta = \dfrac{1}{4}$ and $\vartheta = \dfrac{ln(2)}{ln(3)}$ respectively.

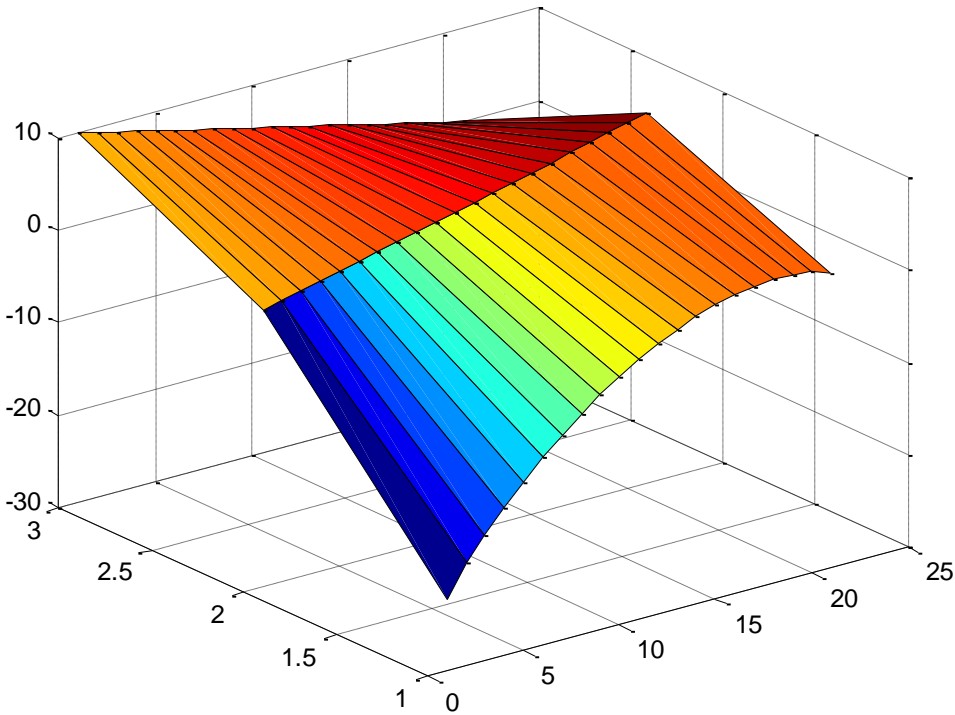

**Figure 1.** The plot of solution to dissipative wave equation with local fractional operators with fractal dimension $\vartheta = \dfrac{1}{4}$.

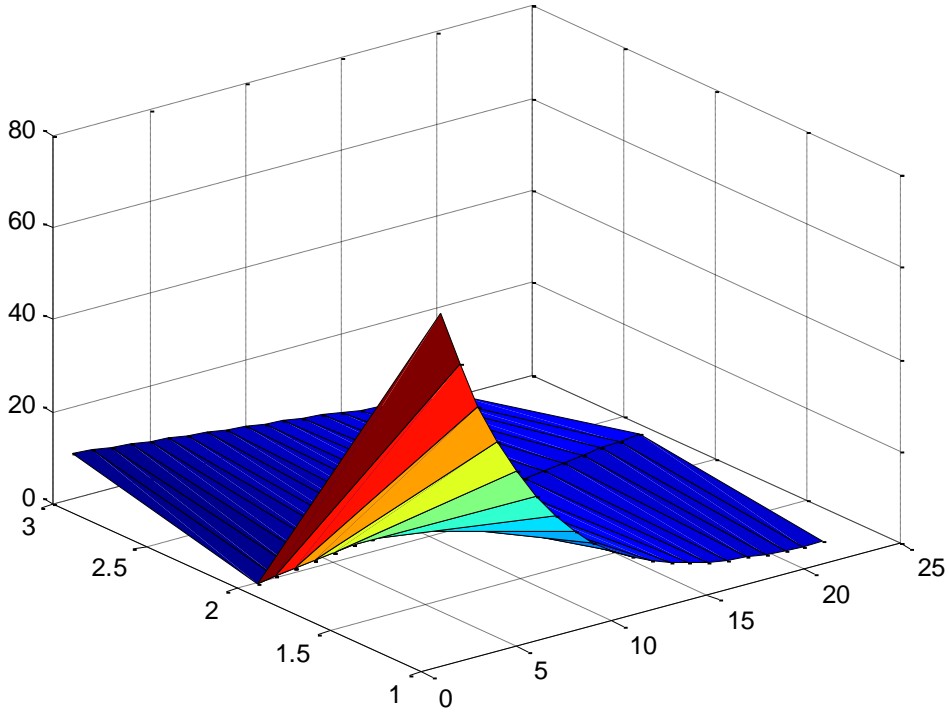

**Figure 2.** The plot of solution to dissipative wave equation with local fractional operators with fractal dimension $\vartheta = \dfrac{ln(2)}{ln(3)}$.

**Example 2.** *Consider the following damped wave equation with LFDOs*

$$\frac{\partial^{2\vartheta}\varphi}{\partial\kappa^{2\vartheta}} - \frac{\partial^{\vartheta}\varphi}{\partial\kappa^{\vartheta}} - \frac{\partial^{2\vartheta}\varphi}{\partial\eta^{2\vartheta}} - \frac{\eta^{\vartheta}}{\Gamma(1+\vartheta)} = 0, \quad 0 < \vartheta \le 1, \tag{32}$$

*with the initialvalue condition as follows:*

$$\varphi(\eta, 0) = 0, \quad \frac{\partial^\vartheta \varphi(\eta, 0)}{\partial \kappa^\vartheta} = -\frac{\eta^\vartheta}{\Gamma(1 + \vartheta)}. \tag{33}$$

*In view of (13) and (32):*

$$
\begin{aligned}
\widetilde{L}_\vartheta \left\{ \varphi_{n+1} \right\} &= \widetilde{L}_\vartheta \left\{ \varphi_n \right\} - \frac{1}{s^{2\vartheta}} \widetilde{L}_\vartheta \left\{ \frac{\partial^{2\vartheta} \varphi_n}{\partial \kappa^{2\vartheta}} - \frac{\partial^\vartheta \varphi_n}{\partial \kappa^\vartheta} - \frac{\partial^{2\vartheta} \varphi_n}{\partial \eta^{2\vartheta}} - \frac{\eta^\vartheta}{\Gamma(1 + \vartheta)} \right\} \\
&= \frac{1}{s^\vartheta} \varphi_n(\eta, 0) + \frac{1}{s^{2\vartheta}} \varphi_n^{(\vartheta)}(\eta, 0) + \frac{1}{s^{3\vartheta}} \frac{\eta^\vartheta}{\Gamma(1 + \vartheta)} + \\
&\quad \frac{1}{s^{2\vartheta}} \widetilde{L}_\vartheta \left\{ \frac{\partial^{2\vartheta} \varphi_n(\eta, \kappa)}{\partial \eta^{2\vartheta}} + \frac{\partial^\vartheta \varphi_n(\eta, \kappa)}{\partial \kappa^\vartheta} \right\}.
\end{aligned}
\tag{34}
$$

*The initial value reads:*

$$\varphi_0 = -\frac{\eta^\vartheta}{\Gamma(1 + \vartheta)} \frac{\kappa^\vartheta}{\Gamma(1 + \vartheta)}. \tag{35}$$

*Hence, we get the first approximation, namely:*

$$\widetilde{L}_\vartheta \left\{ \varphi_1 \right\} = -\frac{1}{s^{2\vartheta}} \frac{\eta^\vartheta}{\Gamma(1 + \vartheta)}.$$

*Thus, we have*

$$\varphi_1(\eta, \kappa) = -\frac{\eta^\vartheta}{\Gamma(1 + \vartheta)} \frac{\kappa^\vartheta}{\Gamma(1 + \vartheta)}.$$

*The second approximations reads:*

$$\widetilde{L}_\vartheta \left\{ \varphi_2(\eta, \kappa) \right\} = -\frac{1}{s^{2\vartheta}} \frac{\eta^\vartheta}{\Gamma(1 + \vartheta)}.$$

$$\widetilde{L}_\vartheta \left\{ \varphi_n(\eta, \kappa) \right\} = -\frac{1}{s^{2\vartheta}} \frac{\eta^\vartheta}{\Gamma(1 + \vartheta)},$$

*and*

$$\varphi_n(\eta, \kappa) = -\frac{\eta^\vartheta}{\Gamma(1 + \vartheta)} \frac{\kappa^\vartheta}{\Gamma(1 + \vartheta)}.$$

*Hence, the solution is:*

$$\varphi(\eta, \kappa) = -\frac{\eta^\vartheta}{\Gamma(1 + \vartheta)} \frac{\kappa^\vartheta}{\Gamma(1 + \vartheta)}, \tag{36}$$

*which is exactly the same as that obtained by LFVIM [21].*

Now, we solve problem (32) by using the LFLDM. From (22), (23), (32) and (33) the iteration algorithm can be written as follows:

$$\varphi_0(\eta, \kappa) = -\frac{\eta^\vartheta}{\Gamma(1 + \vartheta)} \frac{\kappa^\vartheta}{\Gamma(1 + \vartheta)}, \tag{37}$$

$$\varphi_{n+1}(\eta, \kappa) = \widetilde{L}_\vartheta^{-1} \left[ \frac{1}{s^{2\vartheta} - s^\vartheta} \widetilde{L}_\vartheta \left\{ \frac{\partial^{2\vartheta} \varphi_n(\eta, \kappa)}{\partial \eta^{2\vartheta}} \right\} \right]. \tag{38}$$

*Therefore, from* (37) *and* (38) *we give the components as follows:*

$$\varphi_1(\eta, \kappa) = \widetilde{L}_\vartheta^{-1}\left[\frac{1}{s^{2\vartheta} - s^\vartheta}\widetilde{L}_\vartheta\left\{\frac{\partial^{2\vartheta}\varphi_0(\eta, \kappa)}{\partial\eta^{2\vartheta}}\right\}\right] = 0,$$

$$\varphi_2(\eta, \kappa) = \widetilde{L}_\vartheta^{-1}\left[\frac{1}{s^{2\vartheta} - s^\vartheta}\widetilde{L}_\vartheta\left\{\frac{\partial^{2\vartheta}\varphi_1(\eta, \kappa)}{\partial\eta^{2\vartheta}}\right\}\right] = 0,$$

$$\varphi_3(\eta, \kappa) = \widetilde{L}_\vartheta^{-1}\left[\frac{1}{s^{2\vartheta} - s^\vartheta}\widetilde{L}_\vartheta\left\{\frac{\partial^{2\vartheta}\varphi_2(\eta, \kappa)}{\partial\eta^{2\vartheta}}\right\}\right] = 0,$$

$$\vdots$$

*Consequently, we obtain*

$$\varphi = -\frac{\eta^\vartheta}{\Gamma(1 + \vartheta)}\frac{\kappa^\vartheta}{\Gamma(1 + \vartheta)}, \tag{39}$$

*which is exactly the same as that obtained by LFLVIM and LFVIM [21].*

In Figures 3 and 4, the 3-dimensional plots of the approximate solutions of (32) with initial condition (33) are shown for different values of $\vartheta = \frac{1}{4}$ and $\vartheta = \frac{ln(2)}{ln(3)}$ respectively.

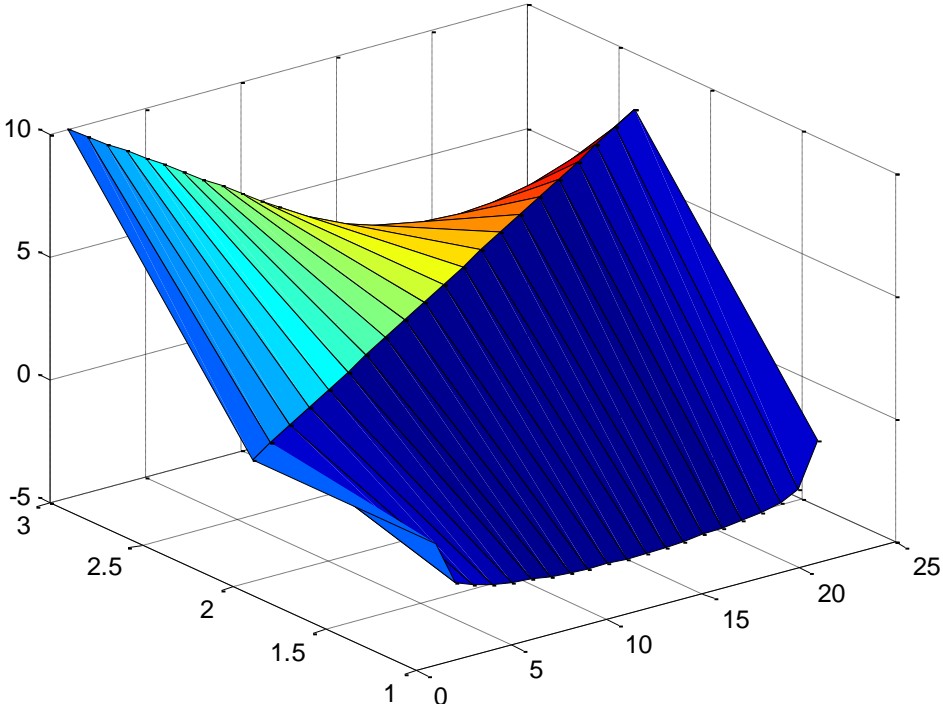

**Figure 3.** The plot of solution to dissipative wave equation with local fractional operators with fractal dimension $\vartheta = \frac{1}{4}$.

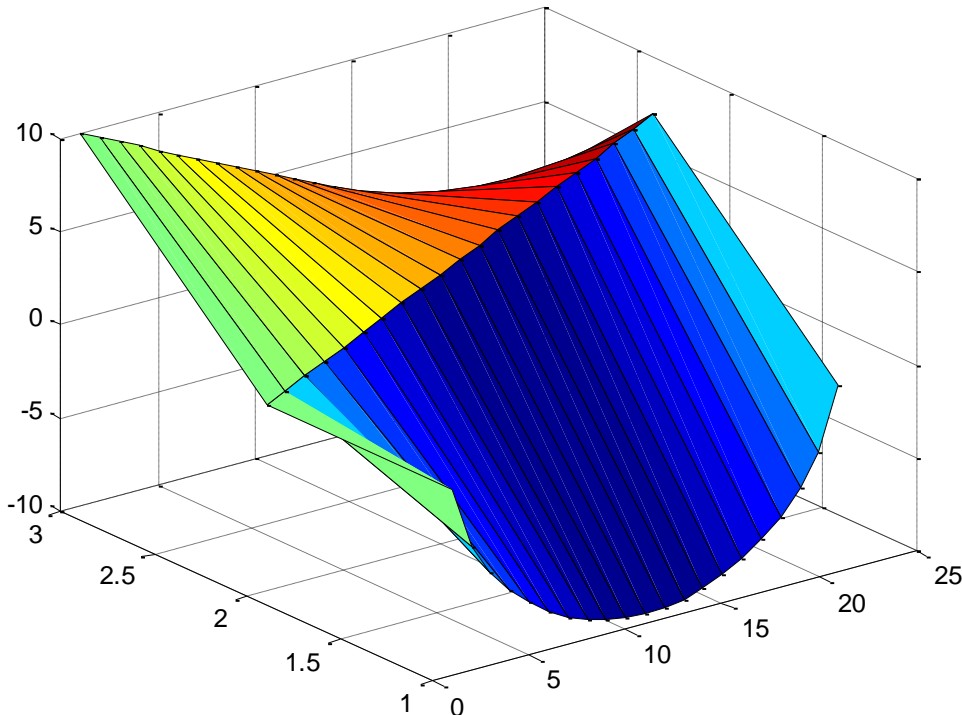

**Figure 4.** The plot of solution to dissipative wave equation with local fractional operators with fractal dimension $\vartheta = \dfrac{ln(2)}{ln(3)}$.

## 5. Conclusions

The LFLVIM and LFLDM have been successfully applied to finding the approximate analytical solutions for dissipative wave equation and damped wave equation with LFDOs. In comparison with local fractional variational iteration method and local fractional Adomian decomposition method, these methods give analytical approximate solutions in series form which converges rapidly. The reliability and the reduction in the size of computational work is certainly a sign of a wider applicability of the methods.

**Author Contributions:** H.K.J. wrote some sections of the manuscript; D.B. prepared some other sections of the paper and analyzed. All authors have read and approved the final manuscript.

**Funding:** This research received no external funding.

**Acknowledgments:** The authors are very grateful to the referees and the Editor for useful comments and suggestions towards the improvement of this paper.

**Conflicts of Interest:** The authors declare no conflict of interest.

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
