# Peer review of "Approximate Solutions of the Damped Wave Equation and Dissipative Wave Equation in Fractal Strings"

_fractalfract, doi:10.3390/fractalfract3020026_

Round 1

Reviewer 1 Report

Comments and Suggestions for Authors

The work is very interesting and has a subject of great relevance in the present day. On the one hand, we have the Damped Wave Equation, on the other hand we have the local  fractional Laplace variational iteration method.

I really enjoyed the work. His reading brings novelty, and information that holds the reader's attention. But, I have some suggestion and considerations to improvement of this work.

Some minor modifications are required, I listed below:

(1) Make clear, the contour conditions. Because in problems with strings we have well defined (and different) conditions.

(2) After all the equation, there is a comma or a period mark. Please, verify all equations (ex: eq. 13).
(3)  what is fractal string? Make mention in the text.

(4)  Finally, I encourage the authors to improve the conclusion.

Thank you for your attention.

Author Response

Reply to the Referee’s Comments

((Manuscript ID fractalfract-495243))

We thank the referees for their valuable comments. Here follows detailed reply to the comments.

Review 1.

v  The conditions are different in the two examples mentioned in the article ((please see (25) and (33) )).

v  All equations were checked in the revised version.

v  A fractal string was added to the introduction.

v  the conclusion is improved in the revised version.

Sincerely Yours,

Dumitru Baleanu

Hassan Kamil

Reviewer 2 Report

Fractal-Frac 495243

The article describes approximate solutions for solving damped wave equation and dissipative wave equation within local fractional derivative operators. However, in order to improve the manuscript, some issues has have to be corrected, as written below:

1 – The Introduction needs to be improved, it’s too short. Put a description of what will be presented in next sections, writing about what are the two examples.

2 – A comparison between what the classical theory applied to the solution of that sort of problems could give the readers a good insight of what is different in the contribution, when you put it together with the examples chosen.

3 – The conclusions are too short again, and needs to be more elaborated.

4 – There are some blank spaces left in the article.

Author Response

Reply to the Referee’s Comments

((Manuscript ID fractalfract-495243))

Dear Prof. Cody Peng,

We thank the referees for their valuable comments. Here follows detailed reply to the comments.

Review 2.

v  the introduction is improved in the revised version.

v  The local fractional Laplace variational iteration method provides us with a convenient way to find solution with less computation as compared with local fractional variational iteration method, as well as local fractional Laplace decomposition method with respect to local fractional decomposition method ((In the conclusion)).

v  the conclusion is improved in the revised version.

v  are some blank spaces have been processed.

Sincerely Yours,

Dumitru Baleanu

Hassan Kamil

Round 2

Reviewer 2 Report

No further recomendations by this reviewer.